# New Species-Specific Primers for Molecular Diagnosis of *Bactrocera minax* and *Bactrocera tsuneonis* (Diptera: Tephritidae) in China Based on DNA Barcodes

**DOI:** 10.3390/insects10120447

**Published:** 2019-12-12

**Authors:** Linyu Zheng, Yue Zhang, Wenzhao Yang, Yiying Zeng, Fan Jiang, Yujia Qin, Jiafeng Zhang, Zhaochun Jiang, Wenzhao Hu, Dijin Guo, Jia Wan, Zihua Zhao, Lijun Liu, Zhihong Li

**Affiliations:** 1Department of Entomology, College of Plant Protection, China Agricultural University, Beijing 100193, China; zlinyu210@163.com (L.Z.); zhangyuejacky@yeah.net (Y.Z.); yangwz96@163.com (W.Y.); zengyiying1996@163.com (Y.Z.); zhzhao@cau.edu.cn (Z.Z.); ljliu@cau.edu.cn (L.L.); 2Institute of Plant Quarantine, Chinese Academy of Inspection and Quarantine, Beijing 100176, China; 13426369960@163.com (F.J.); qinyujia@cau.edu.cn (Y.Q.); 3Hunan Plant Protection and Plant Quarantine Station, Changsha 410006, China; zhbao804@21cn.com; 4Guizhou Plant Protection and Plant Quarantine Station, Guiyang 550001, China; zbjzc@163.com; 5Chongqing Plant Protection and Plant Quarantine Station, Yubei 401123, China; yidao2003090012@163.com; 6Sichuan Plant Protection and Plant Quarantine Station, Chengdu 610041, China; guodijin2008@aliyun.com (D.G.); wan527jia@163.com (J.W.)

**Keywords:** *Bactrocera minax*, *Bactrocera tsuneonis*, molecular identification, species-specific marker, DNA barcodes

## Abstract

Tephritidae fruit flies (Diptera: Tephritidae) are regarded as important damage-causing species due to their ability to cause great economic losses in fruit and vegetable crops. *Bactrocera minax* and *Bactrocera tsuneonis* are two sibling species of the subgenus *Tetradacus* of *Bactrocera* that are distributed across a limited area of China, but have caused serious impacts. They share similar morphological characteristics. These characteristics can only be observed in the female adult individuals. The differences between them cannot be observed in preimaginal stages. Thus, it is difficult to distinguish them in preimaginal stages morphologically. In this study, we used molecular diagnostic methods based on cytochrome c oxidase subunit I and species-specific markers to identify these two species and improve upon the false-positive results of previous species-detection primers. DNA barcode sequences were obtained from 900 individuals of *B. minax* and 63 individuals of *B. tsuneonis*. Based on these 658 bp DNA barcode sequences of the cytochrome c oxidase subunit I gene, we successfully designed the species-specific primers for *B. minax* and *B. tsuneonis*. The size of the *B. minax* specific fragment was 422 bp and the size of the *B. tsuneonis* specific fragment was 456 bp. A series of PCR trials ensured the specificity of these two pairs of primers. Sensitivity assay results demonstrated that the detection limit for the DNA template concentration was 0.1~1 ng/μL for these two species. In this study, we established a more reliable, rapid, and low-cost molecular identification method for all life stages of *B. minax* and *B. tsuneonis*. Species-specific PCR can be applied in plant quarantine, monitoring and control of *B. minax* and *B. tsuneonis*.

## 1. Introduction

Tephritidae, consisting of the true fruit files, is one of the largest families in Diptera. Tephritidae includes over 5000 species classified into 500 genera [1,2], and is distributed worldwide except in Antarctica [1]. The members of the genus *Bactrocera* of Tephritidae, distributed primarily in tropical Asia, Australia and South Pacific regions, are regarded as damage-causing species due to their devastation of fruit and vegetable crops [1]. *Bactrocera minax* (Chinese citrus fly) and *Bactrocera tsuneonis* (Japanese orange fly) are two sibling species of subgenus *Tetradacus* of *Bactrocera* that are native to the East Asia region. At present, there are two theories on the origin of *B. minax*. The first theory indicates that *B. minax* was first detected in Jiangjin County, Sichuan Province (now Jiangjin District, Chongqing City) in 1943. The second theory demonstrates that *B. minax* was detected in Guizhou in the Ming dynasty. Both theories suggest that *B. minax* is native to China [3]. A paper published in 1956 demonstrated that *B. tsuneonis* is native to the southern part of Japan [4]. The above two species are considered to be major pests of citrus crops [5,6], causing large-scale economic losses, doing great harm to the export of citrus, suppressing international trades, and leading to trade barriers. In 2008, there was an outbreak of a serious epidemic in the orangeries of Wangcang County, Guangyuan City, Sichuan Province, where countless *B. minax* individuals devastated the normal production of citrus, causing an enormous panic among consumers, and the citrus market shrank rapidly nationwide [7]. According to many studies on *B. minax* and *B. tsuneonis*, we found that the host range of the two species is different. Although the two species are considered to be major pests of citrus crops, they have different preferences to the species of citrus. *B. minax* has been recorded from *Citrus aurantium*, *C. lemon*, *C. maxium*, *C. medica*, *C. paradist*, *C. reticulata*, *C. sarcodactylis*, *C. sinensis*, *C. tingerina*, *C. unshiu*, *Fortunella X crassifolia*, and *Poncinus trifoliata* [8,9,10]. While *B. tsuneonis* has been recorded from *C. kinokuni*, *C. deliciosa*, *C. unshiu*, and *Fortunella japonica* [11,12]. *B. minax* has a wider host range and this species can infest both thin skin and thick skin citrus species. However, *B. tsuneonis* can only infest thin skin species. This difference might due to the distinction of ovipositors of *B. minax* and *B. tsuneonis*. *B. minax* has a longer ovipositor than that of *B. tsuneonis*.

*B. minax* and *B. tsuneonis* are both univoltine insects. Adult females have a large and powerful ovipositor to lay eggs through the skin of unripe citrus. When the eggs hatch, the larvae eat the inside of the host citrus. Infested fruit will be precocious, yellow early and caduceus, seriously affecting the quality and yield of citrus [13,14,15]. *B. minax* is widely distributed in countries including China, Bhutan, and India [16]. Since it was first detected in Sichuan Province in the 1940s [3], *B. minax* has spread rapidly in China and is reported to be distributed in Guangxi, Guizhou, Hubei, Hunan, Shaanxi, Sichuan and Yunnan Provinces. In contrast, *B. tsuneonis* is limited to China (Guangxi, Hunan, Sichuan), Japan (Ōita-ken, Miyazaki-ken, Kagoshima, Kumamoto-ken, Amami-Ō-shima) and Vietnam [17]. According to recent surveillance data, researchers have firstly trapped *B. tsuneonis* in an orchard (24.3000° N, 123.7200° E) located 30 km northwest of Huaiji County, in northwestern Guangdong Province, China, on 29 April 2016, which suggests that it is spreading in China [18]. At present, there are still no demonstrated lures for the specific trapping of *B. minax* and *B. tsuneonis.* We therefore cannot carry out effective monitoring for these two species. They cannot be mass reared in the laboratory for three reasons: first, *B. minax* and *B. tsuneonis* would diapause in winter [19]; second, they do not prefer an artificial diet; and third, they cannot lay eggs after mating indoors. The above reasons have limited physiological, biological, and biochemical studies and integrated management of *B. minax* and *B. tsuneonis*.

The morphological characteristics used for distinguished *B. minax* and *B. tsuneonis* can only be observed in the adult stage. Moreover, these characteristics are usually present in female adults, which creates great difficulties in identifying the male adult. Intercepted samples are generally in early developmental stages (e.g., eggs and larvae), for which morphological keys are lacking [20]. In different sites, the plantation structures are different. The two species have different geographic distributions, which could be due to different host needs. Inability to distinguish them may lead to their invasion. For example, in thick skinned citrus species planting regions, if someone identified *B. minax* as *B. tsuneonis* and did not take measures to control it, it would infest the fruits and cause huge economic losses.

Normally, the intercepted immature samples are reared in the laboratory over a period of 7–8 months [21] and identified by using adult taxonomic characteristics, which seriously affects the efficiency of plant quarantine. In this case, more reliable methods need to be established.

Molecular diagnostic methods enable precise, reliable identification of immature life stages and infesting males, down to the species level [22]. In 2003, Hebert indicated that the divergence values of DNA barcode sequences between species are generally greater than 3%. In fact, when this value was employed as a threshold for species diagnosis, it led to the recognition of 196 out of the 200 (98%) species recognized through a prior morphological study [23]. However, the limits among species are very variable. Based on DNA barcode sequences, a clear intraspecific threshold was 2.5% for herbivorous fly identification [24], while that threshold was between 5.6–6.0% for dwelling water mite species identification [25]. Renaud et al. (2012) assessed COI sequences of some species of Diptera and indicated the mean pairwise intraspecific distances (range 0.17–1.20%) and maximum intraspecific distances (range 3.00–5.40%) [26]. A 658 base-pair region of cytochrome c oxidase subunit I (COI) is used as a DNA barcode, and this standard sequence provides the taxonomic resolution to discriminate closely related species. In this case, DNA barcodes were employed as an efficient tool to identify species. DNA barcoding based on COI gene sequencing has recently become an effective tool for insect identification. This method has been applied successfully for the identification of fruit flies. DNA barcoding has made a great contribution to plant quarantine approaches. Armstrong et al. (2005) analyzed the COI sequences of fruit fly samples intercepted from ports of New Zealand over the past decade, and the identification results were consistent with previous results from restriction fragment length polymorphisms (RFLPs). In addition, identification by using DNA barcodes identified species that could not be identified by RFLP analysis [27]. Buahom et al. applied the method of DNA barcoding to amplify the sequence of the COI gene from five larvae fruit flies collected from guava fruit from Thailand in 2011, resulting in identification of the flies as *B. correcta* [28]. Barr et al. (2017) used 539 DNA sequence records from 74 species of *Anastrepha* and demonstrated that these barcode data could distinguish four plant pests: *Anastrepha grandis*, *A. ludens*, *A. serpentina* and *A. striata* [29]. Manger et al. (2018) genetically characterized 10 fruit fly species of the genus *Bactrocera* by using the standard DNA barcoding region of the COI gene, and the identification of eight species was straightforward [30]. However, the limitations of this method are also evident; for example, it is time-consuming and expensive.

Based on the COI gene, several rapid diagnostic methods have been developed, such as the PCR-RFLP, real-time PCR, loop-mediated isothermal amplification (LAMP) and microfluidic dynamic array techniques. These approaches have been applied successfully in the detection of multiple economically important fruit fly species.

These approaches have been applied successfully in the detection of some economically important fruit fly species. There have been several studies on the application of PCR-RFLP in fruit fly identification. For example, Onah et al. (2017) successfully applied PCR-RFLP based on the COI gene to identify *B. dorsalis* and *Ceratitis anonae* (Graham) infesting citrus in southeastern Nigeria [31]. Raquin et al. (2018) also developed a PCR-RFLP assay for identifying *Drosophila melanogaster* among field-collected larvae, and they discriminated *D. melanogaster* from other ecologically relevant species of *Drosophila* at the larval stage. The target sequence was a COI gene sequence [32]. Recent studies have also used real-time PCR to identify economically important fruit flies. Li et al. (2019) performed real-time PCR assays for the rapid detection of *Zeugodacus cucumis* and *B. jarvisi* in New Zealand, and although it was difficult to identify them in the immature stage, real-time PCR successfully distinguished *Z. cucumis* from *B. jarvisi* [33]. The advanced method of LAMP is also used to distinguish economic fruit fly species. Blaser et al. (2018) established the LAMP method for providing reliable differentiation between tested regular and nonregular insect species, including some economically important fruit fly species [20]. In addition to these methods, microfluidic dynamic array analysis is a new method that has broad potential to become one of international standards for use in plant quarantine and invasive species detection. Jiang et al. (2016) established a standardized reaction system for detecting economically important tephritid species in six genera (*Anastrepha, Bactrocera*, *Carpomya*, *Ceratitis*, *Dacus*, and *Rhagoletis*) based on a microfluidic dynamic array [34].

The main objective of this study was to establish a rapid, economical and precise molecular diagnostic method based on DNA barcodes and species-specific PCR to distinguish *B. minax* from *B. tsuneonis*. The number of applications for species-specific PCR is not small. Chua et al. (2010) successfully designed two pairs of species-specific primers based on a 1517 bp sequence of the mtDNA COI gene that could distinguish *B. papayae* and *B. carambolae* under the conditions of regular PCR assays [35]. Asokan et al. (2011) designed species-specific primers based on DNA barcode sequences to establish a method for *B. dorsalis* and *B. zonata* identification, and this method could reliably identify all life stages of the target species [36]. It is worth noting that in 2014, Jiang et al. designed species-specific primers of *B. minax* and *B. tsuneonis* based on the DNA barcoding sequences from Guizhou, Sichuan and Hunan Provinces [37]. We found false-positive results when we used these primers for the identification of the two *Tetradacus* species collected from Guizhou, Sichuan, Yunnan, Hubei, and Chongqing Provinces or city areas, which were not involved in the previous study. The design of more reliable and specific primer pairs is urgently needed for their diagnoses.

In this study, 963 samples were collected from 44 locations from eight provinces in China that covered all distribution areas of these two species. Based on the 963 DNA barcode sequences, we redesigned the species-specific primers for *B. minax* and *B. tsuneonis* and tested the specificity of the primers. The results showed that the primers designed in this study were effective in distinguishing these two species. This study could be a robust supplement to the morphological diagnostic method, and our approach could be applied for pest monitoring in the field, providing guidance for pest management and the prevention of their spread.

## 2. Materials and Methods

### 2.1. Sample Collection

A total of 963 individuals of *B. minax* and *B. tsuneonis* were collected from 44 locations in 8 provinces and areas of China (Guizhou, Hunan, Sichuan, Hubei, Shaanxi, Guangxi, Yunnan and Chongqing). Larvae were collected from rotted fruits, and adults were trapped by a mixture of water, sugar, vinegar and white spirits (water:sugar:vinegar:white spirits = 10:4:2:1), a yellow sticky trap or a green trap ball. In total, we obtained 52 adults and 911 larvae. All of the adults were identified using taxonomic keys, and all samples were preserved in 100% ethanol and stored at −20 °C in the Plant Quarantine and Invasion Biology Laboratory of China Agricultural University (CAUPQL) until the forthcoming molecular diagnostic procedures.

### 2.2. Morphological Diagnostic Method

Currently, the main diagnostic method depends on the morphological characteristics of adults [38]. Unfortunately, *B. minax* and *B. tsuneonis* are closely related and are distinguished by only four relatively evident characteristics. Firstly, *B. tsuneonis* has 1–2 pairs of postsutural supra-alar setae, which are absent in *B. minax*. Secondly, *B. minax* has only one pair of scapular setae, and *B. tsuneonis* has two pairs of scapular setae. Thirdly, the oviscape of *B. minax* is comparatively long, at least equal in length to tergites 3–5 as seen directly from the dorsal view; however, that of *B. tsuneonis* is relatively short, less than or equal in length to tergites 5–6 as seen directly from the dorsal view. Finally, the aculeus of *B. minax* is slender and sharply pointed at the apex, whereas that of *B. tsuneonis* is trilobed at the apex [5]. The above characteristics can only be seen in adults or female individuals. A full description of *B. minax* was provided by White and Elson-Harris (1994) [39]. However, the larva of *B. tsuneonis* remains unknown and the characters given may to a large extent also apply to that species, making larval identification unreliable. The two species share similar morphological characteristics in the larva stage, and so these characteristics cannot be used to distinguish these species.

### 2.3. DNA Extraction, PCR Amplification and Sequencing

Total genomic DNA was extracted from a whole larva or the muscular tissue of the adult using the TIANamp Genomic DNA Kit (DP304, TIANGEN, China) following the manufacturer’s protocol for animal tissue. All DNA template concentrations were estimated by spectrophotometry (UV-Vis spectrophotometer Q5000, QUAWELL, USA). The remainder of the DNA was stored at −20 °C as a voucher.

PCR amplifications of the 658 bp mtDNA COI barcode sequences of 963 samples were performed using the universal primers LCO1490/HCO2198 [40]. PCR amplification in a total volume of 25 μL was performed using the following components: 19.5 μL of 2 × Taq PCR MasterMix, 1 μL of forward primer (10 μM), 1 μL of reverse primer (10 μM), 9.5 μL of ddH_2_O and 1 μL of template DNA. The PCR cycling conditions were as follows: initial denaturation at 94 °C for 3 min, followed by 35 cycles of denaturation at 94 °C for 1 min, annealing at 50 °C for 1 min, extension at 72 °C for 1 min, and a final extension at 72 °C for 10 min. Each PCR product (5 μL) was tested by using 1.5% gel electrophoresis in 1 × TAE buffer, and the results were examined under UV light after anthocyanidin staining. PCR products were then sent to Sangon Biotech (Shanghai) Co., Ltd. The DNA barcoding sequence of each individual was identified via the identification module at BOLD (Barcode Of Life Data) or the BLAST strategy at NCBI. Considering the results of several previous studies synthetically, in this study, the clear intraspecific threshold was 3% (similarity > 97%). Sequences were virtually translated to amino acids by MEGA 7.0 software [41] to detect frameshift mutations and nonsense codons, the results show there was no amplified pseudogenes. After that, sequences were deposited in GenBank.

We obtained 963 DNA barcode sequences, 911 DNA barcode sequences were from larvae and 52 DNA barcode sequences were from adults. Distributional information for the two species is presented in Figure 1, and the samples’ detailed location information and GenBank accession numbers of the sequences are provided in Appendix A. These sequences were sequenced by ourselves and uploaded onto GenBank.

### 2.4. Specific Primer Design and Specificity Test

Species-specific primer pairs were manually designed based on variations in COI barcodes. Firstly, the haplotypes of *B. minax* and *B. tsuneonis* were generated by DnaSP v6 [42] for specific primer design. In order to guarantee the reliability of results, some 658 bp standard DNA barcode sequences of various *Bactrocera* species obtained from GenBank were added for specific primer design. Detailed information of sequences downloaded from GenBank is provided in Appendix A. Then, sequence alignment of these haplotypes and standard DNA barcode sequences were performed by using MEGA 7.0 software [41]. C or G sites with intraspecific crosstalk and interspecific variation were selected, and species-specific primers were designed to cover species-specific sites, this process was performed by BioEdit software [43]. Thereafter, one *B. minax* DNA barcode sequence was input into Oligo 7.0 software [44]. The length of the primers was set as 25 bp. Species-specific sites were placed at the 3’-ends of the forward and reverse primers to ensure specificity between the two species. The length between forward and reverse primers was not less than 200 bp. The species-specific primer sequences were checked by Oligo 7.0 software. These sequences were checked according to the following indexes: (a) check the Key info; the absolute value of 3’ ΔG cannot be higher than 9; (b) check the Duplex Formation index; the absolute value of ΔG cannot be higher than 4.5, and the number of binding sites base pairs cannot be higher than 3, for evaluating the forward and reverse primers, respectively; (c) check the Hairpin Formation index; the absolute value of ΔG cannot be higher than 4.5, and the number of binding sites base pairs cannot be higher than 3, for evaluating the forward and reverse primers, respectively; (d) check the Composition and Tm information; GC content should be 40~60%, Tm should between 50~70 °C, and the difference in Tm between the forward and reverse primers should be less than 5 °C; (e) the false priming efficiency should be lower than 100; (f) the length of the primers should be 20~30 bases. The species-specific primers for *B. tsuneonis* were designed in the same manner. The primers were synthesized by Sangon Biotech (Shanghai, China) Co., Ltd.

The specific primers were tested according to the following principles: one species-specific primer could only amplify the specific sequence of its target species, and another eight nontarget species, used as negative controls, showed no evident band when the species-specific primers were matched to the specific sequences of the target species through agarose gel electrophoresis. One individual from each geographical population was selected to carry out the primary specificity test (Appendix A). PCR amplification in a total reaction volume of 50 μL was performed using the following components: 25 μL of 2 × Taq PCR MasterMix, 1.5 μL of forward and reverse primers, 19 μL of ddH_2_O, and 3 μL of template DNA. The PCR cycling conditions were as follows: initial denaturation at 94 °C for 3 min, followed by 30 cycles of denaturation at 94 °C for 1 min, annealing at 65 °C for 1 min, and extension at 65 °C for 1 min, and a final extension at 65 °C for 10 min. Each PCR product (5 μL) was tested by using 1.5% agarose gel electrophoresis in 1 × TAE buffer, and the results were examined under UV light after anthocyanidin staining.

Besides the above experiments, 2–5 individuals from each geographical population were selected for a further specific test. They were amplified with the two optimal species-specific primers respectively for the reliable and accurate verification of the species-specific primers.

### 2.5. Sensitivity Test

*B. minax* and *B. tsuneonis* samples from different locations were tested for the reliable and accurate sensitivity. We selected one individual from each province to perform this test. A dilution series of template DNA was dissolved in ddH_2_O, and the template DNA concentrations were 100 ng/μL, 10 ng/μL, 1 ng/μL, 0.1 ng/μL, 0.01 ng/μL, and 0.001 ng/μL. The PCR amplification conditions were the same as those described above.

## 3. Results and Discussion

### 3.1. DNA Barcode Sequence Analyses

The concentrations of DNA varied from 60~900 ng/μL. Through DnaSP analysis, 51 haplotypes of *B. minax* and 12 haplotypes of *B. tsuneonis* (Appendix A) were obtained. These sequences were used to design species-specific primers. We also downloaded 56 barcode sequences of *B. minax,* 20 barcode sequences of *B. tsuneonis* and barcode sequences of other fruit fly species from GenBank for the primer design (Appendix A). A total of 900 individuals from Hubei, Hunan, Guizhou, Chongqing, Yunnan, and Shaanxi were identified as *B. minax* (similarity from 97.44% to 100%), and 63 individuals from Sichuan, Hunan, Yunnan, Guangxi, and Guizhou were identified as *B. tsuneonis* (similarity from 98.07% to 100%). Taking all the DNA barcode sequences into consideration, 51 haplotypes of *B. minax* and 12 haplotypes of *B. tsuneonis* (Appendix A) were obtained from DnaSP analysis.

### 3.2. Species-Specific Primer Selection, Specificity Test and Sensitivity Test

We firstly used the primers designed by Jiang et al. (Table 1) [38] to test specificity. The agarose gel electrophoresis results of the specificity tests of the previously reported species-specific primers presented nonspecific bands, as shown in Figure 2 and Figure 3. *B. tsuneonis* samples from Sichuan, Guizhou and Yunnan exhibited a false-positive result amplified by using the primers BTmina-F/BTmina-R. *B. minax* samples collected from Hubei, Hunan, Guizhou and Chongqing also presented a false-positive result amplified by using the primers BTtsun-F/BTtsun-R.

Referring to the *B. minax* and *B. tsuneonis* DNA barcode sequences, a pair of *B. minax*-specific primers and a pair of *B. tsuneonis*-specific primers were designed (Table 2). The sites of the species-specific primers in the COI barcode sequences for the target species are presented in Figure 4. The specificity of the species-specific primers was tested by performing PCR assays with the samples listed in Appendix A. We successfully designed specific primers after testing all geographical populations. Products were only amplified for the target species; the lengths of the amplified products were 422 bp and 456 bp respectively; and there were no nonspecific amplicons in any of the other species trials (Figure 5 and Figure 6). The results of the further specificity test are shown in Appendix A.

In a previous study, researchers collected the fruit fly samples only in Guizhou, Sichuan and Hunan provinces. Jiang et al. only obtained three DNA barcode sequences from *B. minax* and two DNA barcode sequences from *B. tsuneonis* [37]. In this case, more reliable species-specific primers were successfully redesigned based on DNA barcode sequences obtained from 963 individuals and download from GenBank with the highest abundance possible, which almost covered all distribution spots of *B. minax* and *B. tsuneonis* in China. We increased the number of samples from Shaanxi, Hubei, Chongqing, Yunnan and Guangxi compared with a previous study. Moreover, we added DNA barcode sequences of other species of *Bactrocera* for designing primers. Thus, we confirmed the specificity of the species-specific primers for *B. minax* and *B. tsuneonis* diagnosis based on DNA barcodes.

To determine the sensitivity of the selected species-specific primers under the above-mentioned PCR conditions, a tenfold dilution series of template DNA was dissolved in double-distilled water from one individual of each species used. The concentrations of DNA are presented in the Materials and Methods. The results showed that the *B. minax* species primers presented evident sensitivity at a concentration of 1 ng/μL for the Shaanxi, Hunan, Guizhou, Yunnan, and Hubei sample DNA templates but presented an evident sensitivity of 0.1 ng/μL for the Chongqing sample DNA template (Appendix A). In contrast, the *B. tsuneonis* species primers presented an evident sensitivity of 1 ng/μL for the Yunnan, Guangxi, and Guizhou sample DNA templates but presented an evident sensitivity of 0.1 ng/μL for the Sichuan, and Hunan sample DNA templates (Appendix A). The sensitivity levels of both the *B. minax* and *B. tsuneonis* species primers were between 0.1~1 ng/μL. However, the *B. tsuneonis* species primers presented a sensitivity of 0.1 ng/μL for two DNA templates. Compared with the *B. minax* species primers, the *B. tsuneonis* species primers provided a higher sensitivity. While the results of the sensitivity test carried out by Jiang et al. demonstrated that the detection limit of the DNA template concentration was 1 ng/μL for *B. minax* and 0.1 ng/μL for *B. tsuneonis* [37]. It was confirmed that the new species-specific primers of these two target species could be used in rather low-DNA-concentration conditions, which increased the feasibility of using these new methods for the identification of *B. minax* and *B. tsuneonis*. Furthermore, the sensitivity of the new species-specific primers was not lower than that of previous primers.

### 3.3. Advantages of This Diagnosis Method

With the development of international trade, potential invasion pests can easily invade new suitable areas via long-distance transport, which could cause serious economic loss and great harm to food security. A rapid and accurate diagnostic method urgently needs to be established and applied to prevent the further spread of such species. At present, although substances such as ammonium acetate and putrescine, or a mixture of water, sugar, vinegar and white spirits in a 10:4:2:1 ratio can be used for the surveillance of fruit flies, there are still no demonstrated lures for the specific trapping of *B. minax* and *B. tsuneonis.* This results in failure of target species detection, further infestation and failure of quarantine identification. Phytosanitary departments should pay more attention to these two species and prevent their further spread and circumvent their establishment in pest free areas.

*B. minax* and *B. tsuneonis* lack pronounced morphological characteristics except the female ovipositor [5]. Therefore, it is difficult to identify these two species by morphological methods. Inaccurate monitoring and identification of *B. minax* and *B. tsuneonis* would lead to pest invasion and serious economic loss in pest-free areas.

Molecular diagnostic methods have been considered an effective method for supplementing morphological methods. In this study, false-positive results of previous studies using *B. minax* and *B. tsuneonis* species-specific primers were discovered when we tested the reliability of previous primers using the samples from new locations that were not involved in the previous study. In this study, the PCR cycling conditions were simplified into two steps, the annealing and extension processes shared the same temperature of 65 °C, in this case, annealing and extension could be regarded as one process. Thus, the process of identification took less time, moreover, the specificity of the trials was increased and the possibility of mis-pairing between the primers and nontarget templates was reduced.

The present assays show several advantages over others assays. Firstly, unlike LAMP methods, the species-specific primers for traditional PCR are more easily designed, and only one pair of primers is needed for each species. In contrast, a set of six primers is needed for each species in LAMP assays, including forward inner primer (FIP) and backward inner primer (BIP) as inner primers, backward outer primer (B3) and forward outer primer (F3) as outer primers, and backward loop primer and forward loop primer as the loop primers [45]. Secondly, the equipment needed for this method includes only an electrophoresis apparatus, PCR instrument and water bath. These instruments are relatively easy to obtain. Thirdly, these methods are cost effective compared with DNA barcoding and save both time and money because no sequencing process is involved, and the PCR assays are simplified. Fourthly, the concentrations of DNA templates needed in the assays are rather low, which increase the practicability of microscale DNA template identification. Finally, the procedures of this methodology can be performed by nonspecialists who have limited knowledge of molecular biology. The samples examined in this study all came from China, and the results represent the Chinese populations’ circumstances.

The results of this study present potential for application in quarantine inspections at ports. Plant quarantine is extremely important for pest management and prevents dangerous pests from spreading from colonized regions to pest free regions. The identification of pests is the pivotal procedure in quarantine measures. According to the correct identification results, proper measures were developed to manage *B. minax* and *B. tsuneonis*, and reduce the losses caused by these species.

## 4. Conclusions

Based on DNA barcode and species-specific markers, we established molecular diagnostic methods for *B. minax* and *B. tsuneonis* in China, corrected previous mistakes and provided a reliable, rapid identification method. To promote the monitoring of dangerous pest and plant quarantine, the present study provides further guidance for the development of control and management strategies for *B. minax* and *B. tsuneonis*. In addition, this study provides further biological study of *B. minax* and *B. tsuneonis.*

## Figures and Tables

**Figure 1 insects-10-00447-f001:**
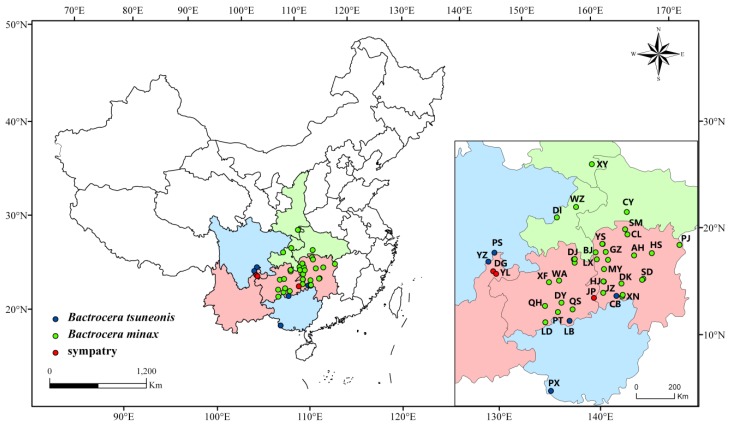
Distributional information for the two species, where each code represents one geographical population, detailed information is provided in Appendix A. The green spots represent the distribution of *B. minax* samples, and the blue spots represent the distribution of *B. tsuneonis* samples. The red spots represent areas of sympatry.

**Figure 2 insects-10-00447-f002:**
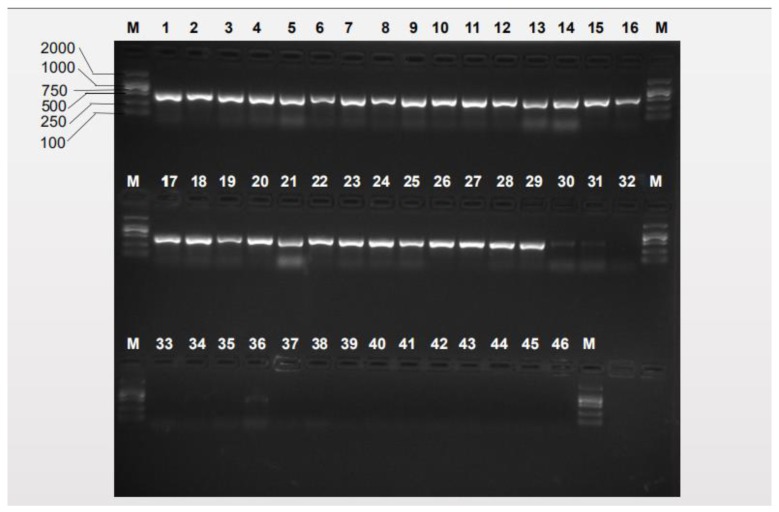
Specificity of the previous Bmina-F/Bmina-R *B. minax*-specific primer pair Lanes 1–29: *B. minax* from 29 geographical populations (Appendix A); lanes 30–38: *B. tsuneonis* from nine geographical populations (Appendix A); lane 39: *B. correcta*, lane 40: *B. dorsalis*; lane 41: *B. latifrons*; lane 42: *B. tryoni*; lane 43: *B. zonata*; lane 44: *Zeugodacus cucurbitae*; lane 45: *Z. scutellatus*; lane 46: *Z. tau*; lane M: D2000.

**Figure 3 insects-10-00447-f003:**
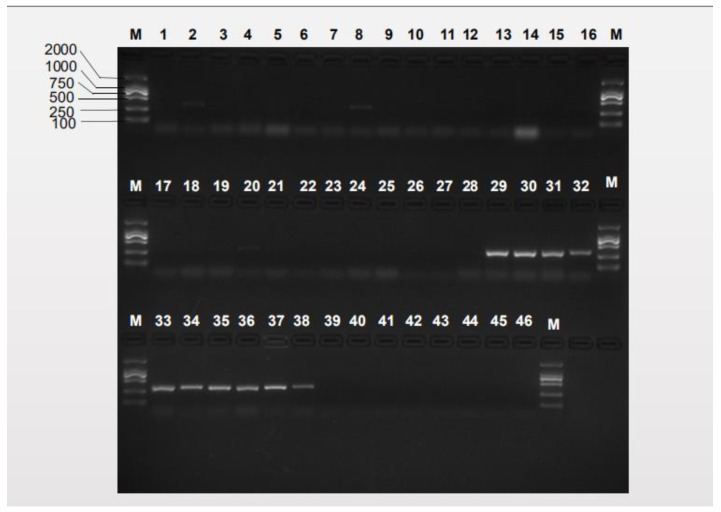
Specificity of the previous Btsun-F/Btsun-R *B. minax*-specific primer pair Lanes 1–29: *B. minax* from 29 geographical populations (Appendix A); lanes 30–38: *B. tsuneonis* from nine geographical populations (Appendix A); lane 39: *B. correcta*, lane 40: *B. dorsalis*; lane 41: *B. latifrons*; lane 42: *B. tryoni*; lane 43: *B. zonata*; lane 44: *Zeugodacus cucurbitae*; lane 45: *Z. scutellatus*; lane 46: *Z. tau*; lane M: D2000.

**Figure 4 insects-10-00447-f004:**
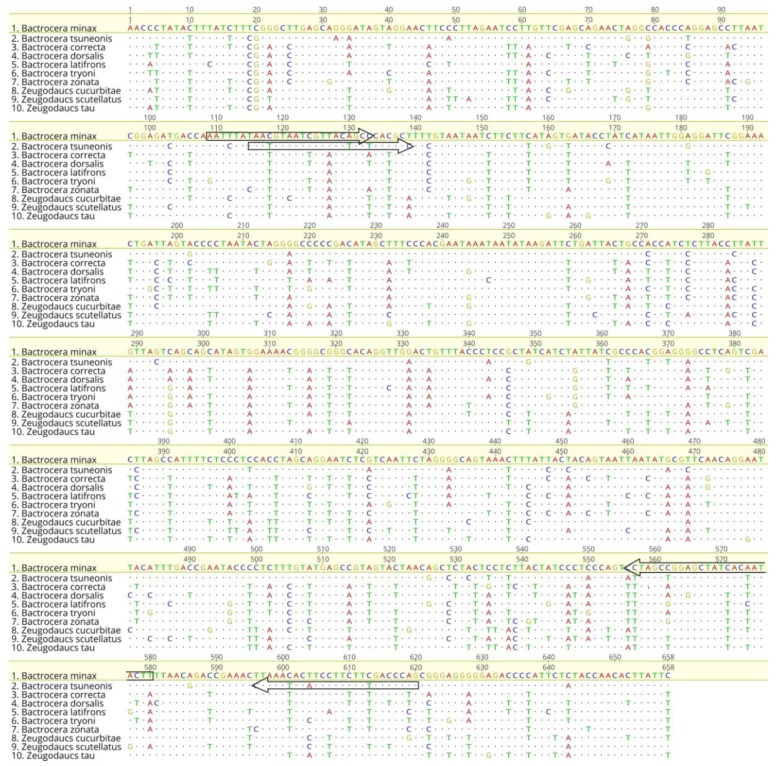
Alignment of COI sequences of 10 species of quarantined fruit flies in China. The sequences indicated by the arrows are the sites of the *B. minax* and *B. tsuneonis*-specific primers in the DNA barcode sequences of the target species. The arrow is in the 5’-3’ direction.

**Figure 5 insects-10-00447-f005:**
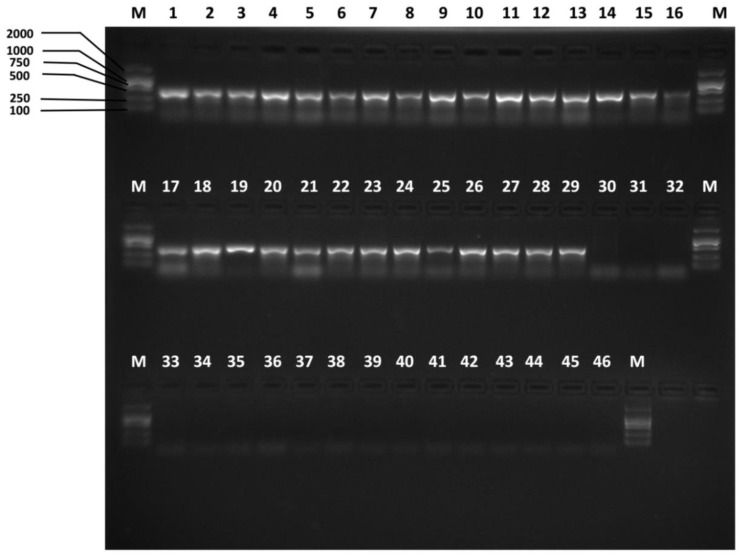
Specificity of the Bm-F/Bm-R *B. minax*-specific primer pair. Lanes 1–29: *B. minax* from 29 geographical populations (Appendix A); lanes 30–38: *B. tsuneonis* from nine geographical populations (Appendix A); lane 39: *B. correcta*, lane 40: *B. dorsalis*; lane 41: *B. latifrons*; lane 42: *B. tryoni*; lane 43: *B. zonata*; lane 44: *Zeugodacus cucurbitae*; lane 45: *Z. scutellatus*; lane 46: *Z. tau*; lane M: D2000.

**Figure 6 insects-10-00447-f006:**
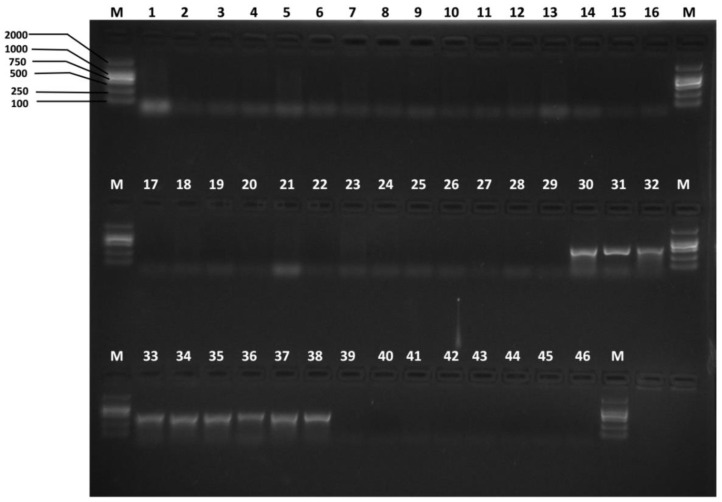
Specificity of the Bt-F/Bt-R *B. tsuneonis*-specific primer pair. Lanes 1–29: *B. minax* from 29 geographical populations (Appendix A); lanes 30–38: *B. tsuneonis* from nine geographical populations (Appendix A); lane 39: *B. correcta*, lane 40: *B. dorsalis*; lane 41: *B. latifrons*; lane 42: *B. tryoni*; lane 43: *B. zonata*; lane 44: *Zeugodacus cucurbitae*; lane 45: *Z. scutellatus*; lane 46: *Z. tau*; lane M: D2000.

**Table 1 insects-10-00447-t001:** List of specific primer sequences for *B. minax* and *B. tsuneonis* designed by Jiang et al. (2014).

Species	Primer	Primers Sequence (5’-3’)	Size (bp)
*B. minax*	BTmina-F	CTTGTTCGAGCAGAACTAGGC	499
	BTmina-R	GGACTGGGAGGGATAGTAAGAGG	
*B. tsuneonis*	BTtsun-F	CCATCCCTTACCCTATTGTTACTC	337
	BTtsun-R	AGGATGTATTTAGGTTTCGGTCC	

**Table 2 insects-10-00447-t002:** List of specific primer sequences for *B. minax* and *B. tsuneonis.*

Species	Primer	Primers Sequence (5’-3’)	Size (bp)	Tm (°C)
*B. minax*	Bm-F	AATTTATAACGTAATCGTTACAGCC	422	53.9
	Bm-R	AAGTATTGTGATAGCTCCGGCTAGG		60.2
*B. tsuneonis*	Bt-F	TAATGTAATCGTTACTGCTCACGCC	456	59.9
	Bt-R	CTGGGTCAAAGAAGGATGTATTTAG		56.1

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
