# Peer review of "New Species-Specific Primers for Molecular Diagnosis of Bactrocera minax and Bactrocera tsuneonis (Diptera: Tephritidae) in China Based on DNA Barcodes"

_insects, 2019, doi:10.3390/insects10120447_

Round 1
Reviewer 1 Report
In this article the authors develop new primer sets to differentiate between two morphologically similar fruit fly species, of Bactrocera minax and Bactocera tsuneonis. The authors present important details concerning the relevance of this primer design as well as provide their barcode sequences for future effort in monitoring these pests. While the article is well written, I was unable to find any supplementary tables (lines 168, 191, 215-216, 228, 264, and 267). The supplementary information provided, consists of a word document containing 48 supplementary figures. The authors need to address this miscommunication by added the tables for clarifying that the supplemental information is in the form of figures and not tables. Overall, the findings are supported by thorough experiments and add important methodologies to the field. In addition to clarifying the supplemental data, I suggest the authors make the following suggested changes to strengthen this article.
Major edits:
The introduction needs to be reorganized to allow the reader to more clearly understand the provided information. Paragraph 1 (lines 40-48) explains the impact of these pests, while paragraph 2 (lines 49-65) seems to expound on (1) distribution (lines 49-57), (2) life cycle (lines 57-62), and (3) impact as it relates for a 2008 outbreak (lines 62-65). I suggest the authors add the outbreak information to the end of paragraph 1, and reorganize paragraph 2, so that the lifecycle information comes before the distribution information. The authors may also consider moving lines 285-299 from the Discussion to the Introduction, as this information provides further support for the experiments completed in this manuscript. I am unclear as to why so much time is spent explaining BOLD (lines 79-86). While I understand the importance of barcoding, the authors expound on BOLD, but do not return to this system at any other time in the manuscript. I suggest making this BOLD section smaller to allow focus to center on examples of the applications of DNA barcoding (lines 86-98). Rename figures (figure legend and in-text) in the order in which they are first mentioned in the manuscript. Italicize all scientific names in the References section.
Minor edits/questions:
22-24: Reword sentence to read “These two species share many morphological characteristics which can only be used to differentiate between species in adult individuals…”
26: Change “corrected” to “improved upon”.
25: Mention that the gene used for barcoding is cytochrome c oxidase here, first.
52: Change “…is distributed in limited areas” to “is limited to”.
54-57: Shorten and combine these sentences, as they provide information concerning the expansion of B. tsuneonis.
58: Change “incubate” to “develop”.
59: Is it necessary to explain the color change in fruit after describing how the fruit rots, as well?
61: Add “has” before “limited”.
66-67: Combines these sentences.
66-67: Consider moving this information to the end of the distribution section of the introduction. This change will allow the reader to see the importance of proper monitoring techniques in the sense of pest expansion.
75: Add “(COI)” after “cytochrome oxidase subunit 1”.
89: Write out the unabbreviated form of RFLP before “RFLP”, and put “RFLP” in parentheses.
92: Add a period after “Thailand” and capitalize “they”.
105: Write out the unabbreviated form of LAMP before “LAMP”, and put “LAMP” in parentheses.
106: Change “some” to “multiple”.
132-133: Explain the ratio of each component of the adult traps.
133-134: Confirm the number of adults and larvae, as these numbers are flipped at line 154.
155: Add a space between “Figure” and “1”.
165: Please provide the version of DnaSP used along with the web address or specific citation.
218: Add the citation for Jiang et al.
220: Rename figures to be in correct order.
226: Rename figures to be in correct order.
231: Rename figures to be in correct order.
275-284: Combines these paragraphs into one larger paragraph.
293-293: Expound on which life stages/sexes can be used to identify these species.
309: Remove “massive”.
311-314: Please explain which methodology is being compared to the new technique.
320: Add a space between “primers” and “[33]”.
329-334: Compare this methodology to the other data concerning fruit flies. Emphasize how your data have added to the protocols available for differentiation of pest species.
Reviewer 2 Report
I have read with attention the MS titled New species-specific primers for molecular diagnosis of Bactrocera minax and Bactrocera tsuneonis (Diptera: Tephritidae) in China based on massive DNA barcodes and authored by Linyu Zheng, Yue Zhang, Wenzhao Yang, Yiying Zeng, Fan Jiang, Yujia Qin, Jiafeng Zhang, Zhaochun Jiang, Wenzhao Hu, Dijin Guo, Jia Wan, Zihua Zhao, Lijun Liu and Zhihong Li. Below you will find my remarks, comments, and suggestions. The manuscript reports the results of the development of a molecular identification method that distinguishes two neighboring species. The method is based on the use of specific primes, this method replaces the specific primers developed previously from other authors which have proved to be unreliable.
General observations:
The aims of paper were not clearly identified, the authors did not clarify why it is necessary to distinguish precisely these two species because both are univoltine and infest the same hosts so that a possible bad identification would not cause any problem for control.
Moreover, using a morphological approach to distinguish adult females is possible and easy as stated also by authors.
I have doubts that this methodology developed has an actual utility and not only scientific.
I found the manuscript difficult to read and very confused I think it is necessary a deep revision of the manuscript to make it more readable, moreover, I am not sure of the usefulness of specific primers to identify individuals of unknown species.
Moreover, the primers were tested only on 8 different species while the potential invasive tephritids species is very high. For this reason, I believe that the manuscript is of very limited importance.
Abstract
Lines 22-24
This sentence should be rearranged because the information is correct but the fact that their morphological features can only be used to identify the adult individuals does not make more difficult to distinguish them morphologically. Maybe something should be added about the difficulty to distinguish the preimaginal stages of these species.
Line 26 I believe “corrected” should be “correct”.
Introduction
Lines 44-46 Authors should give a reference that confirms these species are native from East Asia Region.
Lines 55-57 This sentence has to be rearranged because it is not clear.
Line 58 please change “incubate” with “develop”.
Lines 60-62 Authors should explain why the species cannot be reared in the laboratory. The fact that they are univoltine does not make impossible the rearing. Besides, the authors should add some references about these statements.
Line 69 Please change “in” with “by”
Lines 69-70 this sentence is questionable because the authors did not clarify which difference in the behaviour or in the biology exists between the two species. At the moment because both species are univoltine and infest orange which is the problem if somebody does not distinguish them?
The authors should stress better this argument because this is strongly linked to the aims of the paper.
Lines 72-75 This information is correctly reported but successively several authors showed that the limits are very variable. Please change this sentence by citing more recent papers on the delimitation of species. If authors need, I can give several.
Lines 75-76 Please rearrange this sentence because it is questionable, unfortunately, the sequencing of COI is it is not always sufficient to distinguish species, in particular, the Barcoding fragment that is often not very variable as authors wrote just after some sentence.
Introduction should be improved also eliminating some useless parts (lines 79-86).
Line 87 In the reference lacks the year of publication, as well as in all the references cited in this manner please check if it is correct.
Lines 90-93 Please delete this sentence because I think thousands of researchers each day search the sequence they amplified into Blast or bold to identify insects or another organism.
Line 102 Bactrocera invadens was synonymized with B. dorsalis please rearrange this sentence.
Materials and Methods
Authors have to add the taxonomic keys they used to identify the adults.
Authors should clarify if they used the whole insect or just a piece of it, moreover, authors should clarify if the extraction of DNA was conservative.
Line 153 please delete this sentence because this information is present several times in the ms.
Line 167-170 Please rearrange this sentence because it is not clear.
Lines 176-179please check cannot higher because I think that a verb lacks, moreover, this part is very confused please rearrange.
Line 186 please delete “the” before principles
Line 190 authors wrote: One individual 189 from each geographical population was selected to test specificity
But just some lines below authors write: Amplification was performed for 2-5 individuals from each geographical population.
Please clarify which is the correct information.
I found the paragraph: “2.3. Specific primer design and specificity test” very difficult to read it should be completely rearranged.
Results
Lines 213-214 Authors should clarify better this concept, where did they download sequences from?
I think also the following paragraph is confusing and unclear:”3.1. DNA barcode sequence analyses”.
Line 218-223 This sentence must be rearranged because it is not clear and it is too long (six lines).
Figures 2-4 are after figures 5 please correct the order.
I found the whole paragraph “results and discussion” very difficult to read. I think a deep rearrangement is necessary.
Line 206 Authors wrote: 3. Results and Discussion paragraph, but then they inserted also the paragraph discussion. Please check this apparent incongruence.
Line 282 please change “infection” with “infestation”
Line 304. Please check the spelling of “baecode”.
Authors did not write if sequences were virtually translated to amino acids to detect frameshift mutations and nonsense codons. It must be done to exclude they amplified pseudogenes.
Reviewer 3 Report
In this study, the authors developed new species-specific PCR primers for the detection of Bactrocera tsuneonis and Bactrocera minax. In doing so, the authors provide a reliable technique for the accurate detection of these two species, which should significantly help monitoring efforts of these highly invasive pests. I congratulate the authors for improving the specificity of these primer pairs as compared to ones previously used.
Although I think this study is sound, and may have a positive influence for pest management, especially in China, there are writing mistakes throughout the manuscript. More specifically, mis-spellings and grammatical errors are present. Some sentences are simply unclear. There are mis-statements and a lack of order and consistency within the manuscript.
Below, please find more detailed comments to each section of the manuscript. For the record, I did not go through the references.
Major concerns:
>Title: what do the authors mean by massive DNA barcodes? It seems to me that the barcodes are in the normal size range of general barcodes. i.e. 400-500 bp.
>Supplementary tables are not provided. I have been given access to the supplementary file with the figures but nowhere do I see supplementary tables, which are referred to in the text.
>Figures are out of order. The reference of supplementary information is also out of order. Furthermore, supplementary figures S38-S43 and S44-S48 are referred to in the text on L264. The rest are not mentioned anywhere.
>Inconsistencies and contradictions are found in the “Materials and methods” section. It is important that these are very clear and straight forward. Also, I think the correct references for all software used should be provided.
>There is a “Results and discussions” section along with a separate “Discussions” section. I think that is redundant. I believe this was written by accident as under “Results and discussions” the results are presented but not discussed.
>In results, the specificity of the primers isn’t quite the same within the figures and the text (see below).
>Availability of data and material: The DNA sequences of all but the two main study species are provided here. These sequences have been deposited in previous published work. I think the newly derived sequences from this study should be archived and mentioned here.
Minor concerns:
>L45: Bactrocera tsuneonis mis-spelled.
>L26: False positive results of what exactly could be clarified. e.g. “... false-positive results of previous species-diagnostic/detection methods/primers”.
>L66-70: Few mistakes here, making this paragraph hardly understandable.
e.g. L67: “effective monitor” could be changed to “effective monitoring”
L68: What I think the authors mean is that B. minax and B. Tsuneonis lack morphologically differentiated characteristics except in the female ovipositor?
L69: Rather “... with morphological methods” than “... in morphological methods”.
L69: The last sentence of this paragraph also has a few grammatical errors in it and is not quite clear.
>L90-93: I’d recommend to split this sentence into two.
L>106: A few references could be given as to which fruit fly species have been successfully identified using these rapid approaches.
>L145: Mis-spelling “ddH2”.
>L152: The accession numbers of the sequences deposited to GenBank should be provided in the text or in the data availability section.
>L153: First sentence is a repeat sentence from the previous paragraph. Furthermore, the numbers here are not consistent with the previous paragraph. E.g. in the previous paragraph the authors mention 52 adults and 911 larvae were used and in this sentence 52 larvae and 911 adults are mentioned.
>L157: Supplementary material Table S1 – I did not get access to any supplementary tables.
>Figure 1. If possible, a scale within the inset plot could be provided to better depict the distances between sampling sites. Some readers may find it interesting to see how far apart the allopatric sites are. Also, in figure caption Table S1 is referenced (see above).
>L165-167: I recommend splitting this sentence into two. Then, the authors state that additional sequences from Genbank were used. I think it would be good to state how many additional sequences were included and which ones (accession numbers) in text. Here, the authors refer to Table S2, which I cannot see.
>L165: DnaSP software and BioEdit (L170) should be referenced. Same should be done with all used software in the study.
>L176-185: Writing of this paragraph should be re-checked. Maybe not mandatory, but I would suggest adding a sentence or two explaining the parameters used in Oligo 7.0. software.
>L198: Here, the authors state that 2-5 individuals from each geographical population were used to test the primers’ specificity, but above on L189 the authors state that 1 individual from each geographical population was tested for specificity.
>L202: For the PCR sensitivity test, a little more detail could be provided. E.g. how many individuals were used.
>L217: I find it hard to read and understand the first paragraph of this section (3.2.). Furthermore, the authors mention false positive amplifications, but I do not see this in the agarose gel figure 5 (i.e. I do not see the difference between figure 5 and figure 3; comparison of old and new primer pairs). In figure 6, there appears to be one false positive (well 29) as the authors imply on L223, which is improved using the new primers shown in figure 4.
>L258: It seems to me that the authors simply used dilutions of the original DNA samples to conduct the sensitivity test. Whether this is fine or not I am not sure. It may be worthwhile to measure the DNA concentrations in each sample. Dilutions work in theory but in practice, it is common to have unexpected concentrations after dilutions. Additionally, in my experience, nanodrop, sometimes gives very mis-leading concentrations as opposed to a Qubit instrument. It may also be useful to report how pure the DNA samples are.
>258: This paragraph could be re-written more clearly.
>L279-282: Sentence should be broken down into more sentences. Towards the end of the sentence I find it harder and harder to understand.
>L294: What do the authors mean by “intercepted individuals”. That individuals are usually caught at early developmental stages? This may be fine, not sure if the wording is best suited.
>297: The word “months” is mis-spelled.
>L300-302: First the authors mention that previous methods entailed morphological approaches to species identification, but then in this sentence the authors state previous primers. It seems to me that PCR primers for these species already existed, and did a relatively good job at distinguishing between the two species and that the new primers the authors developed do an even better job.
>L304: “barcode” mis-spelled.
>L305: I think this part should highlight the fact that new primers were developed in this current study as opposed to previous studies. Also, in the materials and methods, the authors mentioned that they used online sources of COI gene sequences in conjunction with new sequences to design primers, this is not mentioned here.
>L309: What do the authors mean by massive DNA barcodes? If I am not mistaken, these are smaller than the ones used previously.
>L327: Sentence should be re-written for clarity.
>L332: Which large scale survey are the authors referring to here? Then, which proper measures were developed to manage these two pest species? Do the authors mean the current study? This could be stated more clearly.
Round 2
Reviewer 2 Report
Please see the file attached
